# Casing Wear and Wear Factors: New Experimental Study and Analysis

**DOI:** 10.3390/ma15196544

**Published:** 2022-09-21

**Authors:** Omer Alnoor Osman, Necar Merah, Mohammed Abdul Samad, Mirza Murtuza Ali Baig, Robello Samuel, Meshari Alshalan, Amjad Alshaarawi

**Affiliations:** 1Department of Mechanical Engineering, King Fahd University of Petroleum and Minerals, Dhahran 31261, Saudi Arabia; 2Interdisciplinary Research Center for Advanced Materials, King Fahd University of Petroleum and Minerals, Dhahran 31261, Saudi Arabia; 3Halliburton, Houston, TX 77072, USA; 4Drilling Technology Team, EXPEC Advanced Research Center, Saudi Aramco, Dhahran 31261, Saudi Arabia

**Keywords:** casing wear factor, friction, wear mechanisms, P110 steel, oil-based mud, water-based mud

## Abstract

To understand and quantify casing wear during drilling operations, an experimental setup with real drill pipe joints (DPJ) and casings was designed and used to carry out wear tests, simulating various operating conditions and environments. P110 steel casing samples were tested under dry and wet conditions. Actual field oil- and water-based fluids were utilized to lubricate the contact area at two different side loads (1000 N and 1400 N) and DPJ speeds (115 and 207 rpm). The results show that for the same testing conditions, the casing wear volume and wear factor under water-based lubrication were more than twice those obtained under oil-based fluid testing. As expected, the wear volume and wear factor were highest under dry conditions. Moreover, it was noticed that, as the normal load was increased at a constant rotational speed (rpm), the wear factor increased. On the other hand, raising the rotational speed at the same applied load reduced the casing wear factor, due to the observed absence of adhesive wear and possible localized softening effects at higher speeds. SEM analyses of the worn areas showed that under dry conditions, the main wear mechanisms were abrasion and delamination. However, both adhesive wear and abrasive wear mechanisms were observed under oil-based lubrication. The energy dispersive spectroscopy (EDS) analysis of the worn surface revealed that at higher loads and speeds, a heavy transfer of particles from the oil-based lubricant took place. On the other hand, some contaminants of the water-based lubricant were observed on the worn surfaces.

## 1. Introduction

The tubular structure of a wellbore can become severely damaged during the life cycle of an oil/gas well. In general, the drill string pipe body might partially or fully touch the casing inner wall under axial loading, particularly at the dogleg section. Therefore, mechanical wear of the casing takes place between the hard surfaces of the tool joint and the inner surface of the casing. [1].

Casing wear may lead to several types of severe drilling accidents, thus compromising the well integrity, which increases the cost of drilling to high levels [2]. Wear is an basic form of material loss; it is categorized as material removal from solid surfaces via mechanical action [3].

The hard-banding of tool joints has been implemented to improve drill-pipe life. However, this increases the casing wear rate and the risk of casing failure, which considerably increase the cost of drilling operations [4].

In general, wear can be classified in terms of the mechanism into abrasive wear, adhesive wear, fatigue wear, spalling wear, and corrosion wear. At specific drill-pipe rotational speeds, the impact wear becomes more severe, especially at the resonating frequencies [5]. All these types of wear mechanism contribute to the overall casing wear at different sections of the well and they must be well predicted, monitored, and/or controlled, since they cannot be avoided. It is always challenging to predict the amount of casing wear, because different factors can affect the amount wear that are not fully understood yet [6].

Previous studies indicated that, besides the side loading, fluid medium, and pipe rotation, some other variables contribute to the amount of casing wear [7,8]. Huimei and Yishan [7] studied casing wear behavior and its relation to the drilling conditions in ultra-deep directional wells. They found that the main drilling factors that affect casing wear are the tool penetration rate, tool rotation speed, mud composition, well geometry, and mud viscosity, as well as the mud density. In addition, the dynamics of the pipe due to severe vibration also will cause casing wear. The whipping and lashing of the tube during lateral vibration results in high dynamic side loading [8]. The variables and combination of variables that have a direct effect on casing wear are the casing sizes, drill-pipe (DP) grade, DP size, hard banding material, applied side forces, and DP rpm [9], as well as drilling fluid type and its additives [10]. For instance, drilling fluids that contain hard particle additives increase the wear amount of the casing. On the other hand, limestone additives have a negligible effect on the amount of casing wear after protective layer formation [11]. A large hole curvature also increases the casing wear rate, especially at the natural frequency [12].

The burst strength of the casing is one of the important mechanical properties that might be degraded as a result of the casing wear, and which can lead to a catastrophic casing failure if the safety factor used in the design of the casing is not sufficient [13]. Therefore, the prediction and analysis of casing wear must be carried out to quantify, measure, and predict the amount of wear [14]. Casing wear factors are essential for the accurate prediction of the amount of casing wear [15]. Accurate wear prediction reduces drilling costs and helps in preventing the casing string catastrophic failures caused by wear [16]. It is also crucial for enhancing well longevity and integrity [17].

Since the wear coefficient is an essential factor that affects the accuracy of casing wear prediction, many researchers have built customized casing wear testing machines to analyze, quantify, and obtain casing wear coefficients under different operating conditions. Sun et al. [11] conducted a full-size casing wear examination with different sizes of tool joint. The results showed that casing wear time and the size of the tool joint had a marginal influence on the nonlinear wear model, contrary to that of the wear-efficiency model. Thus, for a high accuracy of casing wear prediction in complex wells, the nonlinear wear model must be used.

Doering et al [18] used real-sized casing parts and tool joints to study the effects of the tool joint contact pressure on the wear factor and developed an empirical relationship that can predict casing wear. Lian et al [19] also utilized full-size tool joints and three different casing grades, to construct the relationship between the amount of casing wear and the energy dissipation due to internal friction. The goal was to study the effect of rotational speed (RPM), contact load, and casing material grade on the depth of casing wear, as well as to develop a wear prediction model.

Zhang et al. [6] found that casing wear mechanisms can be classified as both abrasive and adhesive wear, and might coexist during drilling operations. However, adhesive wear is dominant under high contact loads, while abrasive wear is dominant if the drilling fluid contains a high percentage of hard weighing agent. Other wear mechanisms such as spalling and checking were also noticed on the tested samples. Moreover, hard-worked layers were developed on the worn surfaces, which means that adhesive wear was the dominant wear mechanism. 

Mao et al. [20] studied the effect of drilling tool rotational speed on casing wear with a pin-on-disc laboratory apparatus. The results showed that the wear rate varied non-linearly with any increase in drilling pipe RPM.

Mao et al. [21] studied the effect of oil to water ratio on the behavior of casing wear, by conducting pin-on-disc tests with drilling fluid. The results revealed that adding diesel oil to drilling fluids reduced casing wear considerably. A reduction of wear rate and wear depth was observed as the ratio of the oil to water increased from 0 to 3.

Casing, liner, and tubing wear is a significant issue that needs to be addressed as more and more intricate wells are drilled. Casing wear is one of the ongoing issues the industry is dealing with, and it is challenging how to accurately predict downhole wear. In the past, several casing wear estimating techniques were used in real drilling settings, but the outcomes of those efforts did not always achieve the required level of wear forecast accuracy. As a result, the industry has faced numerous difficulties with the field applicability of the current wear-estimation techniques. Due to the general overdesign of the casings caused by the heavy use of excessive casing-wear safety factors, the impacts of incorrectly estimating casing wear may not be felt. 

An essential component of casing wear estimate approaches are the casing wear factors. However, when the anticipated casing wear is compared to measured data, the experimental wear factor values do not correspond to field circumstances. So that these calibrated values can be efficiently utilized for more precise forecasts, wear factors are adjusted and modified to fit field data. Casing wear factors are now primarily considered to be “fudge factors”, thanks to these, frequently extensive, field calibrations. As a result, there is a lot of misunderstanding about the basic concepts underlying the use of wear factors for casing wear assessments. 

Based on the current complex drilling conditions, the introduction of newer material types for tool joints and casings calls for a fresh round of experiments using these new materials. In order to understand the wear process and post conditions of the material that had been worn out, the following approach was taken

-Comprehensive casing grade characterization-Continuous temperature monitoring-Detailed surface profiling-Microstructure evaluation of the worn-out piece-Different lubricants between the stationary pipe and the rotating member.

In this paper, a conventional lathe machine was modified and automated, and then connected to a data acquisition system, to conduct tool-joint/casing wear tests under various operating conditions and test environments. The setup uses real-sized casing parts and tool joints, to study the effects of the tool joint contact pressure, rotational speed, and type of mud on the casing wear factors and wear characteristics, and for microstructure evaluation of the sample after the tests.

## 2. Experimental Procedure and Testing Facility

The automated casing wear testing facility is shown in Figure 1. The control system is designed to apply and maintain a constant force between the casing sample and the tool-joint using a feedback loop. The force is measured using a dynamometer Type 9139AA, and the displacement is obtained using a stepper motor coupled to a gearbox.

Circular sections with a width of about 25 mm were cut from actual casing pipes at sixty degrees. The sectioned casing sample was mounted on the casing holder, before mounting it on the dynamometer. Figure 2 illustrates how the casing sample was mounted onto the holder. The rigid holder was designed such that it fits various casing sizes.

During the wear test, several parameters such as casing temperature, radial displacement, rotational speed, normal force, and friction force were measured and recorded. Wear volume was estimated based on the wear track geometry using the 3D optical profilometer. 

Two different side loads (1000 N and 1400 N) were applied using a specially designed drive, and two different real-size DP speeds (115 RPM and 207 RPM) were selected, to investigate their impact on the casing wear. The first test was performed under dry conditions, and all other wear tests were performed under wet conditions (water-based or oil-based drilling fluids), to investigate the effects of drilling fluid on the casing wear. A submersible circulation pump was used to supply the drilling fluid to the contact surface between the casing sample and the drill pipe. A strong magnet was used to collect the debris from the drilling fluid reservoir, as shown in Figure 3. A small hole was drilled in the side of casing samples, near the casing-tool joint contact region, to measure the average temperature using a waterproof DS18B20 digital thermal probe sensor that has an operating temperature range: of −55 °C to +125 °C and accuracy of ±0.5 °C. An electronic digital micron indicator that has a measuring range of 0–12.7 mm, resolution of 0.001 mm, and accuracy of ±0.004 mm was used to measure the radial displacement during the test. After the selected load was applied at the desired rotational speed, the origin (zero point) of the digital indicator was identified and the radial displacement of the casing sample was recorded every 2 min. The duration of each wet wear test was 190 min, while the dry test was stopped after 10 min, because of the high temperature developed near the casing-tool joint interface. 

## 3. Characterization

The hardness values of the received samples were measured using a universal hardness testing machine (INNOVA-model 783-D, 2012). Inductively coupled plasma optical emission spectrometry (ICP-OES PlasmaQuant^®^ PQ 9000) was used to determine the elemental composition of the received samples. To obtain the worn surface profile and measure the worn volume of casing samples, a two-inch length of the casing, which included the worn region, was sectioned. A GT-K Contour 3D optical profilometer (Bruker Nano GmbH) was used to measure the wear track area at different locations. The wear track area was estimated by analyzing both the 2D and 3D profiles of the wear tracks using Vision 64 software (Version 5.6) attached to the optical profilometer. The area was estimated at three different locations on the wear track, and the average value is reported. Subsequently, the total wear volume was calculated by multiplying the average wear track area by the length of the wear track. 

To study the casing wear mechanisms, casing samples were sectioned and then prepared for characterization under a scanning electron microscope (SEM) (JEOL SEM model JSM-6460 with EDS facility and JEOL Gold Sputter model JFC-1100).

## 4. Results and Discussion

### 4.1. Elemental Composition

An optical emission spectrometer was used to determine the elemental composition of the P110 casing material samples and the hardened facing of the drill pipe (DP) material. Each sample was analyzed at four different locations, and the average elemental compositions are reported in Table 1 along with American Petroleum Institute (API SPEC 5CT) standard values for the P110 casing grade. Based on these results, the tested casing material had almost the same composition as the standard P110 API SPEC 5CT grade [22].

### 4.2. Hardness Measurements

A Rockwell Hardness tester was used for hardness measurements on the flat and ground surface of the as-received P110 casing and DP samples. A diamond cone indenter with an indenting load of 150 Kgf was used. An average of 10 measurements of each sample were taken. The hardness (HRC) values for the P110 and drill-pipe hard-facing were found to be 30.84 ± 0.976 and 57.94 ± 0.8, respectively [23].

### 4.3. Casing Wear Volume Estimation

An optical profilometer was used to measure the area of the wear groove on the P110 casing material samples. As mentioned above, the tests were conducted under dry and wet conditions at two different loads (1000 N and 1400 N) and at two speeds (115 and 207 rpm). An example of the 3D and 2D wear groove profiles of sample number (S1) obtained by the 3D optical profilometer is shown in Figure 4. All wear volumes of the tested casing samples, shown in Table 2, were measured using an optical 3D profilometer. The results show that the wear volume increased with the side load and rotational speed. Based on these test results, it was observed that using water-based drilling fluids instead of oil-based drilling fluids negatively affected the casing wear and the wear volume increased by 100%, on average. In general, oil-based drilling fluids are preferred, due to their better lubrication performance at various operating conditions, including high temperature and high pressure, compared to environment-friendly water-based drilling fluids [24].

### 4.4. Casing Wear Factors

A commonly used equation to calculate the specific wear factor (K) in mm^3^/Nm is:(1)$$K=\frac{V}{\mathrm{PL}}$$
where P is the applied normal load, L is the total sliding length, and V is the worn volume. The total sliding length L is equal to π × D × N × t. Where D is the drill-pipe diameter in (mm), N is the drill-pipe rotational speed (rpm), and t is the wear test duration (min). The calculated wear factors for P110 samples, tested under different conditions are shown in Table 3. In general, for water-based drilling fluids with a steel-tool joint, wear factor values are between 0.5 and 40 (10^−10^ psi^−1^) or 7 to 580 (10^−9^ MPa^−1^). However, for oil-based drilling fluids with a steel tool-joint the wear factor values are between 0.3 and 5 (10^−10^ psi ^−1^) or 4 to 73 (10^−9^ MPa^−1^) [25,26]. The present wear factor values were within the lower limits of the above reported normal ranges.

### 4.5. Effects of Drilling Fluids Type on Wear Factor

In general, lubricants are used to reduce friction and temperature [27]. During drilling operations, water-based and oil-based drilling fluids are used to transmit hydraulic power to the downhole, remove wear debris from the wellbore, lubricate and cool down the drilling bit, and stabilize the drilling operation. By looking at the wear factor values shown in Table 3. It can be observed that the wear factor under dry conditions was almost six-times higher than that obtained under oil-based lubrication.

Moreover, the casing wear factor obtained for casing samples tested under water-based conditions was higher than that of the oil-based lubrication, as shown in Figure 5. This was mainly due to the lower viscosity of water-based drilling fluids compared to oil-based drilling fluids, which led to having a higher wear volume and friction coefficient for casing samples tested using water-based drilling fluids [28]. Therefore, oil-based drilling fluids are more popular compared to water-based ones. However, today, water-based drilling fluids are coming back, with the recent global environmental movement, since they have less impact on the environment compared to oil based drilling fluids. 

### 4.6. Effects of Rotational Speed and Side Load on the Wear Factor

Drill-pipe rotational speed (N) is directly proportional to the total sliding distance (L), and both (P) and (L) are used to calculate the wear factor, as shown in Equation (1). To understand the effect of each parameter, several tests were conducted at two different speeds, while maintaining the same applied side load and two different side loads at the same rotational speed. Based on the obtained results illustrated in Figure 6, it is noticed that as the normal load increased at the same rotational speed (rpm), the wear factor increased, due to the increase of the real area of contact [6]. On the other hand, raising the speed to a higher value (from 115 to 207 rpm) at the same applied load (1400 N) reduced the casing wear factor by 76%, since this reduced the friction coefficient [29]. This could be attributed to the increase in the localized temperature, which resulted in softening of the material, leading to easy deformation, because of which the COF was reduced [30].

### 4.7. Casing Wear Mechanisms

The received P110 casing sample S0 was tested under the dry condition at 115 rpm and a 1000 N applied load. The test was conducted for only 40 min, because the temperature reached 125 °C and kept on increasing, which could have damaged the dynamometer. SEM images at different magnifications of the P110 worn casing surface after conducting the wear test under the above conditions are shown in Figure 7. It can be observed that the dominant wear mechanism was abrasive wear, as evidenced by the longitudinal grooves on the worn surface [31] and the plastic flow under the effect of load and shear stress [21].

SEM images of the worn casing sample S1 tested under oil-based lubrication are shown in Figure 8. In general, abrasive wear can be identified by the linear grooves observed on a worn surface. On the other hand, adhesive wear is associated with material detachment of the worn surface. As highlighted by the elliptical orange shape, it can be noticed that both adhesive wear and abrasive wear mechanisms are present, which was also observed by other researchers [18]. Moreover, some solid particles were observed on the worn surface, which caused three-body abrasion wear [32]. It is believed that these solid particles come from the oil-based drilling fluid, as evidenced by the EDS analysis of the worn casing surface shown in Figure 9. It was found that for all drilling fluid compositions, abrasive wear was the main wear mechanism [33]. The compositions of the oil-based and water-based drilling fluids provided by a local oil company are illustrated in Table 4 and Table 5, respectively. The mud density for the oil-based lubricant was 80 pcf and that of the water-based mud was 70 pcf. Carbonates and oxides of Ca, Mg, Si, and Al are the primary constituents of marble. The presence of these elements on the worn casing surface indicated the presence of marble, which is a major constituent of the drilling fluid, as indicated in Table 4. The Ca, O, and Cl could also have been transferred from the lime and calcium chloride present in the oil-based drilling fluid. The sulfur may indicate the presence of barite (BaSO_4_).

The SEM images of the worn casing surface of sample S2, shown in Figure 10, prove that the dominant wear mechanism was three-body abrasive wear. By comparing Figure 8 and Figure 10, it can be noticed that as the normal load decreased from 1400 N to 1000 N at 115 rpm, the abrasive wear became more dominant, and no adhesive wear was observed.

SEM images of the worn surface of Sample S3 are shown in Figure 11. It can be seen that there were no signs of adhesive wear, and abrasive wear was the main wear mechanism. A heavy transfer of particles from the oil-based drilling fluid onto the worn surface was also observed.

To study the effect of drilling fluid type on casing wear mechanism, a fifth test was carried out under water-based wet conditions at 115 rpm and 1400 N. Based on SEM images of the worn casing surface of sample S4, as shown in Figure 12, it is clearly noticed that both adhesive wear and abrasive wear mechanisms are present. Moreover, some contaminants of the water-based drilling fluid were found on the worn casing surface.

## 5. Conclusions

Casing wear is one of the key concerns during the drilling phase and should be carefully monitored and tightly controlled. P110 casing material samples were subjected to dry, oil-, and water-based wear tests, as well as two distinct loads and speeds. The wear groove on the worn surfaces was measured using an optical profilometer.

It was found that the wear volume increased with the contact force and tool joint rotational speed. Additionally, it was found that employing water-based drilling fluids instead of oil-based drilling fluids has a detrimental impact on casing wear and causes a more than 100% increase in wear volume and wear factor.The wear groove depth and the wear volume measurement with the optical profilometer provided a more accurate estimation compared to other previous experiments.Additionally, it was observed that the wear factor increased as the typical load increased at a constant rotating speed (rpm). On the other hand, increasing the speed at the same side load resulted in a lower casing wear factor. This might be explained by the adhesive wear mechanism’s reported absence at higher speeds.

## Figures and Tables

**Figure 1 materials-15-06544-f001:**
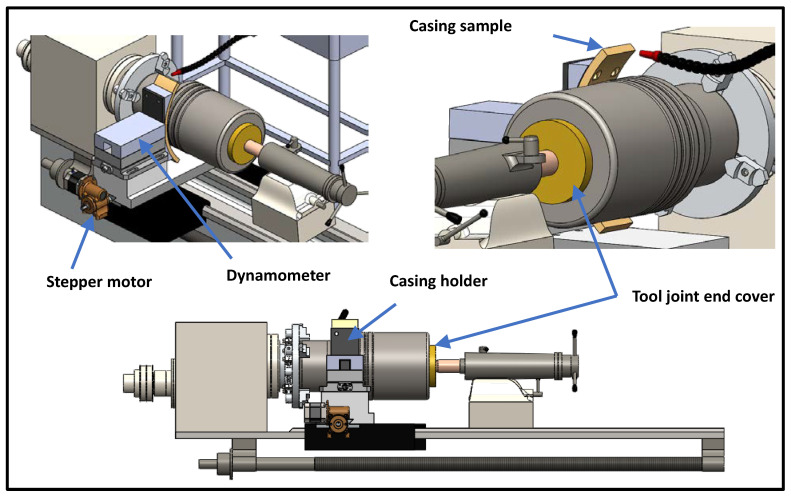
Three-dimensional model of the customized and modified system.

**Figure 2 materials-15-06544-f002:**
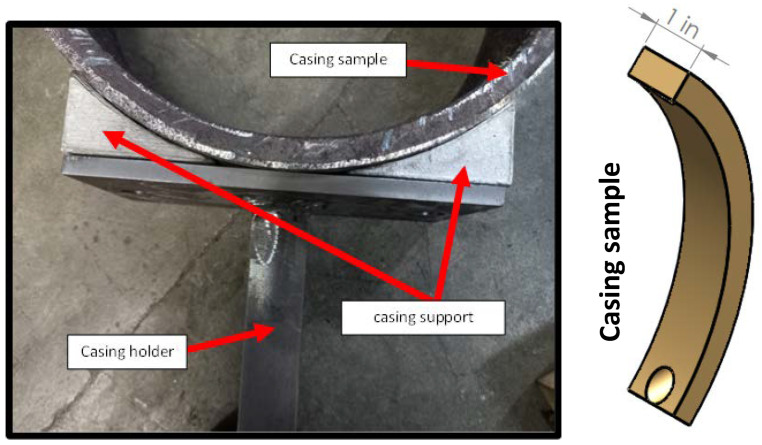
Casing holder and support assembly.

**Figure 3 materials-15-06544-f003:**
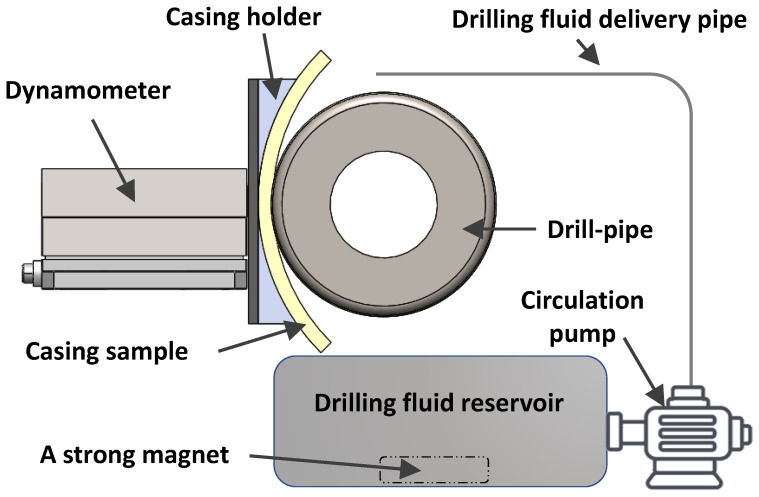
Two-dimensional illustration of the casing wear testing system.

**Figure 4 materials-15-06544-f004:**
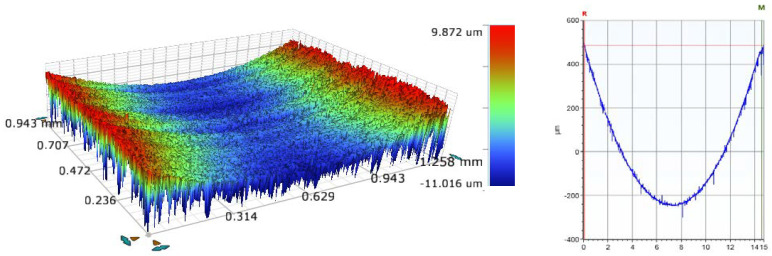
The 3D and 2D wear track profiles of sample number (S1) were obtained using a 3D optical profilometer.

**Figure 5 materials-15-06544-f005:**
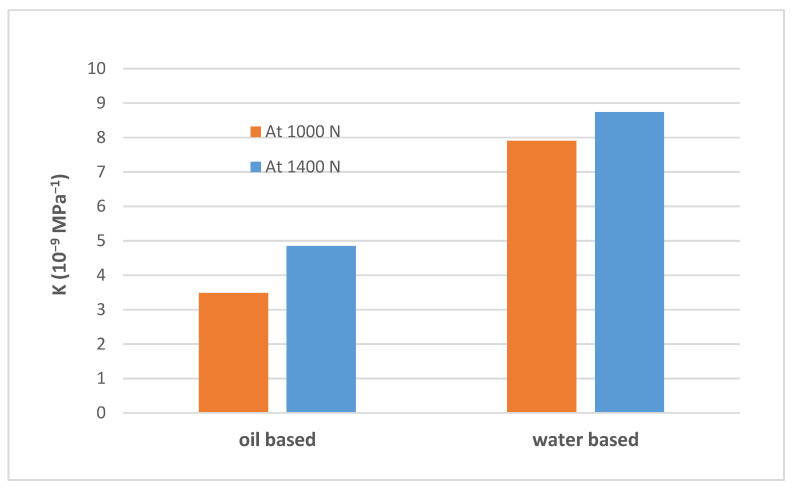
Wear factors of the P110 casing sample test at 115 rpm and two different loads (1400 N and 1000 N) for 190 min under oil-based and water-based wet conditions.

**Figure 6 materials-15-06544-f006:**
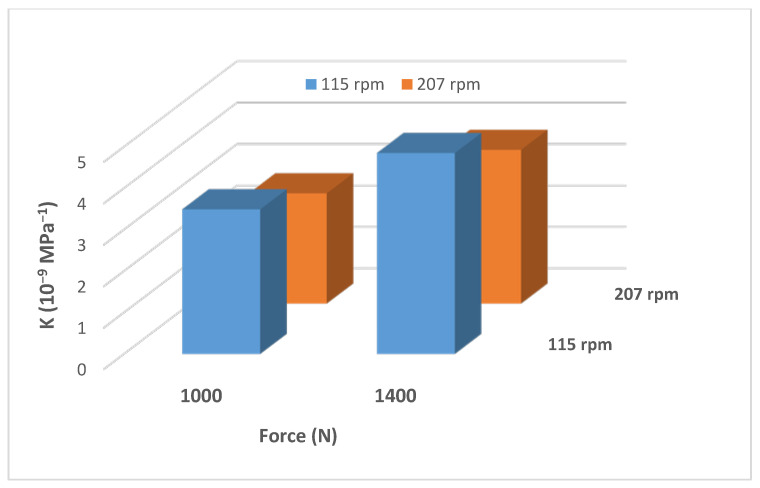
Wear factors of P110 tested under oil-based wet conditions, at two different loads and two different speeds.

**Figure 7 materials-15-06544-f007:**
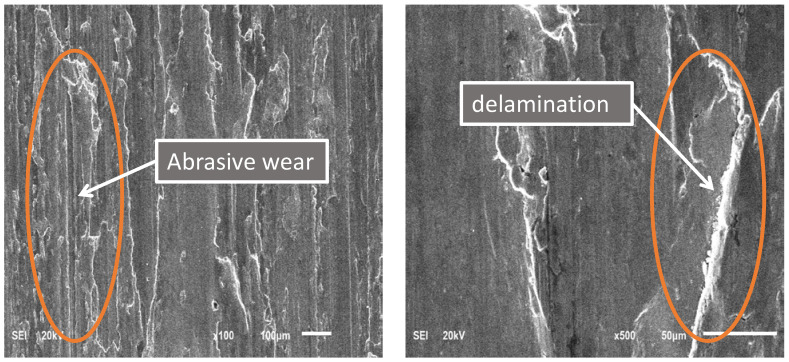
SEM images of the P110 worn casing sample (S0) tested under dry conditions at 115 rpm and 1000 N for 40 min.

**Figure 8 materials-15-06544-f008:**
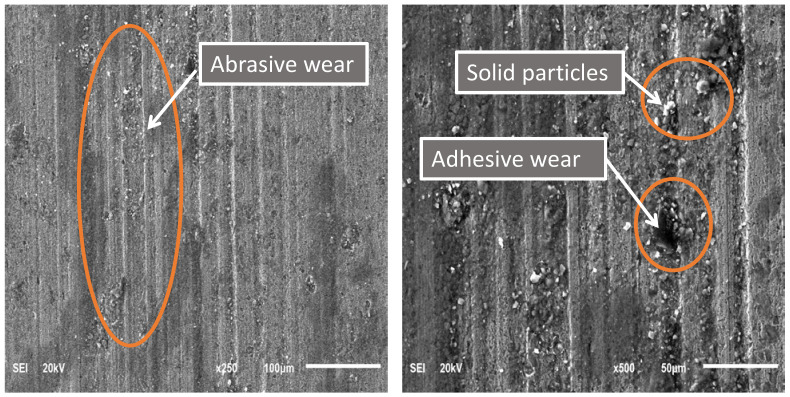
SEM images of a P110 worn casing sample (S1) tested under wet conditions (oil-based drilling fluid) at 115 rpm and 1400 N.

**Figure 9 materials-15-06544-f009:**
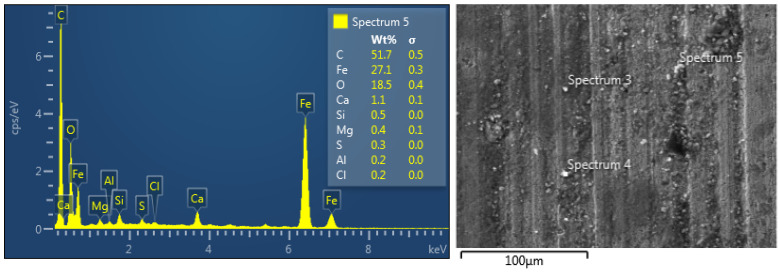
EDS analysis of the worn casing sample (S2) tested under oil-based wet conditions at 115 rpm and 1000 N.

**Figure 10 materials-15-06544-f010:**
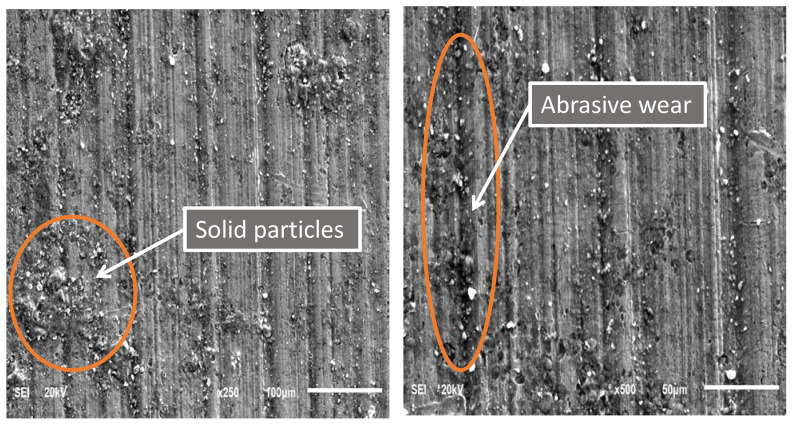
SEM images of a P110 worn casing sample (S2) tested under wet conditions (oil-based drilling fluid) at 115 rpm and 1000 N.

**Figure 11 materials-15-06544-f011:**
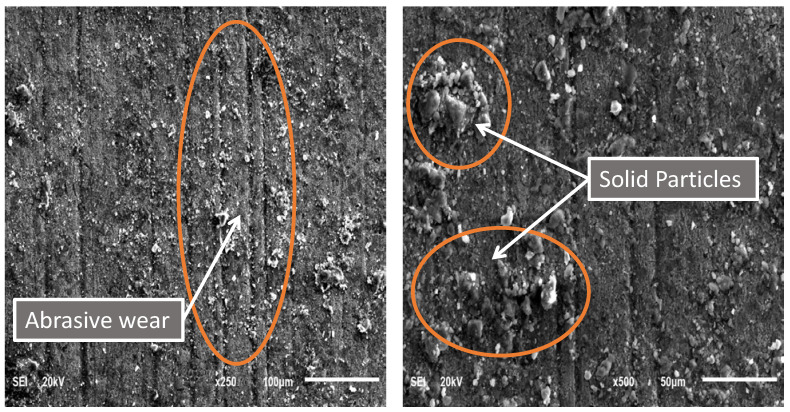
SEM images of the P110 worn casing sample (S3) tested under wet conditions (oil-based drilling fluid) at 207 rpm and 1400 N.

**Figure 12 materials-15-06544-f012:**
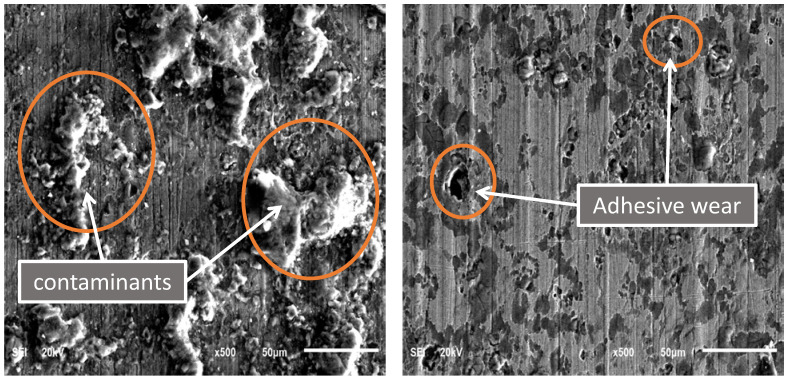
SEM images of the P110 worn casing sample (S4) tested under wet conditions (water-based drilling fluid) at 115 rpm and 1400 N.

**Table 1 materials-15-06544-t001:** Average elemental composition of casing material samples and the hardened facing of the drill pipe material.

Elements	Standard	Measured	Measured
	(P110) API SPEC 5CT	P110	DP, Hardened Surface
C	0.26~0.395%	0.36%	0.96%
Cr	0.80~1.10%	1.17%	3.78%
Ni	≤0.20%	0.04%	0.17%
Mo	0.15~0.25%	0.90%	0.50%
Mn	0.40~0.70%	0.36%	1.34%

**Table 2 materials-15-06544-t002:** Wear volume of the tested casing samples.

Sample No.	Test Condition	Speed (rpm)	Side Load (N)	Average Wear Area (mm^2^)	Wear Groove Width (mm)	Wear Volume (mm^3^)
S0	Dry	115	1000	1.7	23.1	39.3
S1	Oil-based	115	1400	2.3	25.5	59.1
S2		115	1000	1.3	22.5	30.4
S3		207	1400	3.7	22	81.2
S4	Water-based	115	1400	4.5	23.5	106.4
S5		115	1000	3.0	22.7	68.7

**Table 3 materials-15-06544-t003:** The casing wear factor (K) of P110 samples tested under different conditions.

Drilling Fluids Type	Sample Number	F (N)	N (rpm)	D (mm)	T (min)	V $\left({mm}^{3}\right)$	L (mm)	K $\left(MPa^{-1}\right)$	K $\left(psi^{-1}\right)$
Dry	S0	1000	115	126.7	40	39.3	1.831 × 10^6^	21.5 × 10^−9^	1.48 × 10^−10^
Oil-based	S1	1400	115	126.7	190	59.1	8.70 × 10^6^	4.85 × 10^−9^	0.33 × 10^−10^
	S2	1000	115	126.7	190	30.4	8.70 × 10^6^	3.49 × 10^−9^	0.24 × 10^−10^
	S3	1400	207	126.7	190	81.2	15.66 × 10^6^	3.71 × 10^−9^	0.26 × 10^−10^
Water-based	S4	1400	115	126.7	190	106.4	8.70 × 10^6^	8.74 × 10^−9^	0.60 × 10^−10^
	S5	1000	115	126.7	190	68.7	8.70 × 10^6^	7.90 × 10^−9^	0.55 × 10^−10^

**Table 4 materials-15-06544-t004:** The composition of the oil-based drilling fluid, as received from the oil company labs.

Additive Name	From	To	Unit	Function
Safra Oil	0.52	0.51	Barrels	Base oil
Invermul	1.5	1.5	Gallon per Barrel	Emulsifier
Lime_(Ca(OH)_2_)	6	6	Pounds	Contaminant Remover
Duratone	6	8	Pounds	Fluid Loss Control
Fresh Water	0.15	0.15	1/32 inch	
Gel Tone II	6	10	Pounds	OBM Viscosifier
EZ MUL	0.5	0.5	Gallon per Barrel	Emulsifier
Calcium Chloride	33.5	33.1	Pounds	Weighting Materials
Marble Medium	30	30	Pounds	Weighting Materials
Barite	86	100	Pounds	Weighting Materials

**Table 5 materials-15-06544-t005:** The composition of the water-based drilling fluid, as received from the oil company labs.

Additive Name	From	To	Unit	Function
Drill Water	0.85	0.83	Barrels	Base solvent
Soda Ash	0.2	0.5	Pounds per Barrel	Contaminant Remover
Bentonite Baroid	10	10	Pounds per Barrel	Viscosifier
Caustic Soda_(NaOH)	0.5	0.7	Pounds per Barrel	PH Adjustment
Kla Stop	2	3	Volume Percent	Shale Inhibitor
Salt-NaCl	70	80	Pounds per Barrel	Weighting Material
Starch	3	4	Pounds per Barrel	Fluid Loss AdditiveLow Temperature
Xanthan Gum Mi Sch	0.5	1	Pounds per Barrel	Viscosifier
ME LUBE	2	3	Volume Percent	Lubricant

## Data Availability

All supporting data are provided in this paper.

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
