# Peer review of "Casing Wear and Wear Factors: New Experimental Study and Analysis"

_materials, 2022, doi:10.3390/ma15196544_

Round 1

Reviewer 1 Report

1. Please further simplify some of the languages of the abstract. Focus on the research work carried out in this paper and the conclusions drawn.

2.The research on casing wear has been going on for many years, and there are many classical achievements. In the last paragraph of the introduction, please emphasize the innovation of this article and the necessity of carrying out casing wear test of this specification.

3. The density of water-based and oil-based drilling fluids is not described in detail, and the liquid density also has an obvious effect on casing wear, but what is the density of the two kinds of drilling fluids? It is suggested to make a supplementary statement.

4. The conclusion part does not highlight the innovation of the research results of this paper. With the increase of lateral force and rotational speed, the casing wear coefficient increases, and the casing wear coefficient in the air is much higher than that in the fluid, which is generally recognized. It is not necessary to carry out the experiments in this paper to know that. It is suggested to improve the language expression of the conclusion again.

Reviewer 2 Report

The authors presented a detailed experimental study of the tool-joint/casing wear tests and investigated the wear behaviors of P110 casing material under different testing conditions (e.g., dry/wet, rotational speed, load force, etc.). In general, the study is well presented and fits the scope of Materials Journal. However, the authors need to address the following modifications for improvement of manuscript quality. 

  1. 1. The Abstract needs to be well modified to be more concise. The current version is too long and contains too much introduction details and experimental details, which should go to the corresponding sections in the manuscript context body. 

  1. 2. Typos and grammar errors need to be fixed, including but not limited to: Line 14, missing a word after “will severely be”; Line 20, A tool-joint/casing wear tests …; Line 254, missing “of” after “effects”. 

  1. 3. The sentence in Line 115-117 is the same as the one in Line 122-124. 

  1. 4. Line 216, Table 3 should be Table 2. 

  1. 5. Line 248, what do authors mean by “the recent environmental movement”? 

  1. 6. In Line 286-288, the authors indicate the solid particles are from the oil-based drilling fluid per EDS analysis. The authors should clarify this statement by providing details. 

  1. 7. It is very difficult to follow the impact of varying testing conditions on the wear behaviors from the descriptions of SEM findings from the different samples. The authors should provide more clear explanations and discussions of the results. Sample numbers should be included in figure captions for easier follow-up. 

  1. 8. In Conclusions, no conclusions were made on the results from casing wear mechanism study. 

Round 2

Reviewer 2 Report

The authors have addressed the suggested modifications, and the paper quality has been greatly improved.